# Effectiveness of Lyoprotectants in Protein Stabilization During Lyophilization

**DOI:** 10.3390/pharmaceutics16101346

**Published:** 2024-10-21

**Authors:** Vinoothini Karunnanithy, Nur Hazirah Binti Abdul Rahman, Nur Atiqah Haizum Abdullah, Mh Busra Fauzi, Yogeswaran Lokanathan, Angela Ng Min Hwei, Manira Maarof

**Affiliations:** 1Department of Tissue Engineering and Regenerative Medicine, Faculty of Medicine, Universiti Kebangsaan Malaysia, Cheras 56000, Kuala Lumpur, Malaysia; vinoothini8205@gmail.com (V.K.); atiqah.haizum@ukm.edu.my (N.A.H.A.); fauzibusra@ukm.edu.my (M.B.F.); lyoges@ppukm.ukm.edu.my (Y.L.); angela@ppukm.ukm.edu.my (A.N.M.H.); 2Advance Bioactive Materials-Cells UKM Research Group, Universiti Kebangsaan Malaysia, Bangi 43600, Selangor, Malaysia; 3Ageing and Degenerative Disease UKM Research Group, Universiti Kebangsaan Malaysia, Bangi 43600, Selangor, Malaysia

**Keywords:** lyoprotectant, protein stabilization, freeze-drying, cryodesiccation

## Abstract

**Background:** Proteins are commonly used in the healthcare industry to treat various health conditions, and most proteins are sensitive to physical and chemical changes. Lyophilization, also known as freeze-drying, involves sublimating water in the form of ice from a substance at low pressure, forming a freeze-dried powder that increases its shelf life. Extreme pressure and varying temperatures in the freeze-drying process may damage the protein’s structural integrity. Lyoprotectants are commonly used to protect protein conformations. It is important to choose a suitable lyoprotectant to ensure optimal effectiveness. **Method:** Twenty articles screened from Scopus, Web of Science, and PubMed were included in this review that discussed potential lyoprotectants and their effectiveness with different protein models. **Results:** Lyoprotectants were categorized into sugars, polyols, surfactants, and amino acids. Lyoprotectants can exhibit significant protective effects towards proteins, either singularly or in combination with another lyoprotectant. They exert various interactions with the protein to stabilize it, such as hydrogen bonding, hydrophobic interactions, electrostatic interactions, and osmoprotection. **Conclusions:** This review concludes that disaccharides are the most effective lyoprotectants, while other groups of lyoprotectants are best used in combination with other lyoprotectants.

## 1. Introduction

Freeze-drying, also known as lyophilization, is widely used to preserve proteins in a dry state for long-term storage and transportation. These frozen-dried bioproducts facilitate transportation to distant locations without the need for cold-chain storage and reduce point-of-care labor costs by eliminating the need for on-site mixing of reagents. Lyophilized protein powder can be stocked for future outbreaks, and this strategy can even be used in experiments with more extended time scales [1,2].

Lyophilization is an ideal process to convert protein solution into a solid form. The freeze-dried protein powders will increase their shelf-life. However, protein integrity is compromised since the drying process exerts pressure and causes irreversible conformational changes in the protein structure, which leads to the inactivation of the protein functions. During reconstitution of the freeze-dried protein powder, moisture causes the freeze-dried proteins to undergo disulphide interchange and other reactions, which lead to inactivation [1,2]. A FTIR spectroscopy study conducted by Ipsita Roy and team in 2004 exhibited that freeze-drying proteins leads to conformational changes in protein structure. They concluded that the drying stage decreased the α-helix and random structure and increased the β-sheet form in the protein. Freeze-dried fibroblast growth factor and γ-interferon proteins showed significant conformational changes and aggregation during freeze-drying [3]. In order to prevent denaturation and aggregation of protein molecules, lyoprotectants need to be added to stabilize the protein conformation. When aqueous phase stability is a barrier, lyophilization is the most popular method to generate a freeze-dried product. It is crucial to preserve materials that require a sterile and delicate preservation technique and minimal moisture (less than 1%) to maintain stability [4,5]. Freeze-drying is primarily used to safely eliminate moisture from sensitive products, typically those with biological origins, so they can be stored easily, kept in a state suitable for long-term storage, and reactivated by just adding water [1,3]. However, additional stressors from the freeze-drying procedure may result in protein aggregation, fragmentation, and loss of activity. The process usually leads to an increase in the β-sheet formation and a decrease in the α-helix and random structure [3]. It is commonly known that cryoprotectants can protect the protein from denaturation in the early phases, while lyoprotectants are necessary to prevent protein inactivation after drying [6,7]. Adding cryoprotectants alone is not sufficient to ensure protein integrity, as the drying phase can significantly impact protein stability. Therefore, lyoprotectants are required to shield the proteins during the drying phase [3,8].

Lyoprotectants are added to protein formulations to protect them from freeze-drying stresses. They achieve this by reducing the water content and maintaining proper protein structure and stability during the process. The choice of lyoprotectant depends on the protein characteristics, freezing conditions, and desired final product properties. Commonly used lyoprotectants include sugars, polyols, amino acids, and surfactants [9]. Choosing the right lyoprotectant is crucial to preserving the stability and activity of proteins during the freeze-drying process. In protein formulations, sugars, polyols, amino acids, and surfactants are often utilized as lyoprotectants. To create novel and more potent lyoprotectants for protein stabilization during freeze-drying, more investigation is required. This systematic review discusses and identifies the most promising lyoprotectant to preserve protein structure during freeze-drying.

## 2. Methodology

### 2.1. Search Strategy

A comprehensive search was performed in accordance with the Preferred Reporting Items for Systematic Reviews and Meta-Analyses (PRISMA) guidelines. The search was conducted in electronic databases such as PubMed, Web of Science (WOS), and Scopus. To ensure only current studies were used, the search was restricted to publications from 2018 to 2024. Three sets of keyword terms were used for the searching process: “lyoprotectants” or “protective agents” or “chemical additives”; “protein stabilization” or “conformational stability” or “protein activity”; and” lyophilization” or “freeze-drying” or “cryodesiccation”. The title and abstract of all articles were screened individually and duplicate articles were excluded. The references of all retrieved and relevant articles were also reviewed to ensure more thorough findings. The search was restricted to published articles and journal clinical trials in English language. Figure 1 shows the comprehensive search results which were performed according to PRISMA guidelines. The details of search strategy and PRISMA checklist are recorded in the Appendix A [10].

### 2.2. Data Extraction and Synthesis

Two authors (V.K. and N.A.H.A.) independently recorded data from every included study. Any disagreement between the two authors was resolved by discussion with other authors. Then, the following data were extracted: study design, type lyoprotectants, effect on protein models, and alternative freeze-drying method.

The inclusion and exclusion criteria outlined are essential for determining the scope of the systematic review focused on lyoprotectants for protein stabilization during freeze-drying. The review considered studies published between 2018 and 2024. This time frame ensures that the review was based on the most recent and relevant research, reflecting the latest advancements in the field. Only full-text articles published in peer-reviewed scientific journals were included. This criterion was crucial for ensuring the credibility and scientific rigor of the studies being reviewed. This review included articles published in English. This limitation was set to facilitate comprehensive understanding and accessibility of the research findings. This review encompassed studies conducted globally, allowing for a wide range of data and insights from various geographical contexts. Only reviews of published original research articles in indexed journals were included. This criterion ensured that the review was grounded in established scientific evidence rather than anecdotal or unverified claims.

For the exclusion criteria, articles that were not present in indexed journals, such as online articles, review articles, letters to editors, perspectives, commentaries, and news reports, were excluded. This helped maintain the quality and reliability of the sources included in the review. Articles published in languages other than English were not considered. This exclusion was necessary due to the challenges of accurately interpreting and translating scientific content, which could lead to misinterpretation of the findings. By adhering to these inclusion and exclusion criteria, the systematic review aimed to compile a comprehensive and credible body of evidence regarding the effectiveness of lyoprotectants for protein stabilization during the freeze-drying process.

## 3. Results and Discussion

This systematic review was aimed to screen and identify potential lyoprotectants. Discovering new lyoprotectants is important for several reasons, primarily related to enhancing the effectiveness, stability, and safety of the lyophilization (freeze-drying) process, which is widely used for preserving sensitive biological materials, pharmaceuticals, and food products. As formulations become more complex, containing multiple active ingredients or excipients, discovering new lyoprotectants that are compatible with these formulations becomes crucial. Effective lyoprotectants facilitate faster and more complete reconstitution of the lyophilized product back into solution. This is particularly important in clinical settings, where quick preparation is often necessary. As new classes of therapeutics, such as mRNA-based drugs and gene therapies, are developed, they may require unique lyoprotectants to maintain their stability and effectiveness.

The database search yielded 79 results in total. Out of these, 42 articles failed to match the inclusion criteria, and 17 were discarded as duplicates. Ultimately, 20 studies were included in this review. We discovered plausible lyoprotectants from multiple sources that can preserve various protein models during the freeze-drying process after reviewing and contrasting our databases. According to the data, we list the most commonly used lyoprotectant, new potential protein stabilizers, effectiveness of combining two or more lyoprotectants, and interactions between the lyoprotectants and protein. Table 1 summarizes the identified lyoprotectants from the selected articles for further analysis.

### 3.1. Classification of Lyoprotectants

Lyoprotectants can be broadly classified into several categories based on their chemical compositions and functional properties, each serving a specific role in stabilizing proteins during lyophilization. Table 2 classifies the identified lyoprotectant into groups such as monosaccarides, dissacarides, trisaccarides, amino acids, solubilizers, polysaccarides, surfactants, Ions, chemical additives, antiplasticized polymeric glasses, and protein isolates.

### 3.2. Influence of Interaction Between Lyoprotectant and Protein

The important underlying mechanism of lyoprotectant is the interactions between the lyoprotectant and protein. Bonding between lyoprotectant and protein is another key contributor to enhance the stability of the protein conformation in the post-drying process. Hydrogen bonding with proteins is one of the key mechanisms by which lyoprotectants stabilize proteins. Lyoprotectants like sugars (e.g., sucrose, trehalose) and polyols (e.g., glycerol, sorbitol) can form hydrogen bonds with the polar groups on the surface of proteins. This bonding mimics the hydrogen bonding interactions that water would normally provide [31,32,33,34]. By replacing water in these interactions, lyoprotectants help to stabilize the protein structure even in the absence of water. For example, Crowe et al. (1988) demonstrated that trehalose forms a protective glassy matrix that can stabilize dry biological membranes and proteins by replacing the water molecules that normally surround them [35]. The formation of hydrogen bonds between the sugar and protein reduces the protein’s conformational flexibility, thus stabilizing its native structure. Lyoprotectants can interact with hydrophobic regions on protein surfaces, shielding these regions from exposure to the air–water interface during drying. This interaction reduces the tendency of proteins to aggregate, as aggregation often occurs when hydrophobic patches are exposed and interact with one another. Studies, such as those by Wang (2000), suggest that the presence of lyoprotectants can enhance the solubility of proteins and prevent aggregation by covering the hydrophobic regions [2]. This reduces protein–protein interactions that lead to aggregation [36]. Sucrose and trehalose, for instance, have been shown to preferentially exclude themselves from the protein surface, effectively crowding around the protein and stabilizing its native state by shielding hydrophobic patches. Some lyoprotectants can influence the ionic environment around proteins. For example, certain amino acids or ionic compounds may act as lyoprotectants by interacting electrostatically with charged groups on the protein surface, thus stabilizing the protein’s charge distribution and preventing aggregation or denaturation [37,38,39]. Studies have shown that amino acids like arginine can act as stabilizers through electrostatic interactions. They interact with the charged groups on the protein surface, which helps maintain protein solubility and prevents aggregation. For instance, Arakawa et al. (2007) showed that arginine can stabilize proteins by interacting with aromatic and charged residues on the protein surface [40]. These interactions play a critical role in maintaining the native structure of proteins during freeze-drying and storage, thereby preserving their biological activity and preventing degradation. Understanding these mechanisms is vital for developing effective freeze-drying formulations in the pharmaceutical and biotechnology industries [39,41]. Lyoprotectants can interact with proteins in several ways which are summarized in Table 3.

### 3.3. Influence of Glass Transition Temperature Value (Tg′) and Water Replacing Ability of Lyoprotectant on Protein Activity and Strcutural Stability

Tg′ the glass transition temperature value is a vital attribute to consider when choosing an appropriate lyoprotectant during the lyophilization (freeze-drying) process for protein stabilization. Proteins can denature during the freeze-drying process because of dehydration and temperature stress. To avoid that, the product temperature should always remain below Tg′, since exceeding this limit might imply structural changes in the freeze-concentrated amorphous phase which can compromise protein stability. A study by Searles et al. (2001) has demonstrated that in the prevention of structural collapse of freeze-dried cake and denaturation of protein in primary drying, the temperature should be kept below Tg′. The above Tg′ lysozyme and other proteins aggregated so extensively during lyophilization that they lost their activity [7,42].

The freeze-concentrated phase of the ice becomes rubbery rather than less rigid when temperatures exceed Tg′. It leads to total collapse of the product, making its quality and mechanical stability poor in lyophilization. Studies by Tang and Pikal (2004) showed that products lyophilized above Tg′ resulted in marked collapse with poor reconstitution properties and compromised drug quality. The collapse temperature was highly correlated to Tg′, and lyoprotectants that increased Tg′ (sugars) helped to address this challenge. The higher the Tg′ of a formulation, the more likely it is that the protein will be stable during lyophilization. Higher Tg′ formulations perform better in long-term storage and have demonstrated the capacity to withstand temperature cycling [43]. The role of lyoprotectants that increase Tg′ is to trap protein molecules in a rigid glassy matrix, which minimizes molecular mobility and suppresses aggregation as well as oxidation and other degradation mechanisms. Studies demonstrated that lyophilized formulations with higher Tg′ possess better stability in storage as evidenced by reduced protein aggregation and increased biological activity maintenance [44,45].

According to an experiment by Tobias and team, disaccharides-derived polyols such isomalt, lactitol monohydrate, and maltitol have an acquired amorphous state, high Tg′, and are non-reducing because they lack a carbonyl group. These lyoprotectants performed consistently well against 12 model proteins, successfully preserving the protein’s integrity [21]. Disaccharides can lower the water activity in a solution, making it less conducive to chemical reactions that might lead to protein degradation. This low water activity can help preserve the stability of proteins over time as trehalose possess antioxidant properties and can scavenge free radicals that might otherwise damage protein structures. Sugars, being less polar, can shield the hydrophobic regions of proteins from water, thus stabilizing the protein structure. Monosaccharides act as osmolytes, helping to regulate osmotic stress on the biological material [6,7,46]. Maltotriitol and melezitose displayed good protein stabilization, amorphous nature, and are acceptable to good cakes, making them an effective lyoprotectants. Moreover, a study was conducted to observe the lyoprotection and stabilization of laccase extract from *Coriolus hirsutus* using selected additives and found that mannitol was stable at various storage conditions, and the reconstituted enzyme was stable at different incubation temperatures. The results also indicated that adding mannitol to the reaction medium increased the enzyme’s thermal stability [23]. Irreversible aggregation upon freeze–thaw cycles was observed on lyophilized Rituximab antibody with mannitol as a stabilizer [26]. In a study of lyophilizing the protein myoglobin with sucrose and trehalose, trehalose was found to be more preferentially hydrated by water than sucrose. Trehalose forms less hydrogen bonds with the protein surface than sucrose. The rotational motion around trehalose’s two glucose rings is slower than around sucrose’s glucose and fructose rings which results in a less disrupted protein structure in the case of trehalose [18]. Table 4 summarizes the identified lyoprotectants with Tg′ value and their general eligibility as lyoprotectants.

The selection of a lyoprotectant for a protein comprised several key requirements such Tg′ value, water replacing ability, ability to maintain the structural stability, and ability to enhance reconstitution [31,32]. The Tg′ of a lyoprotectant plays a crucial role in determining the stability of proteins after freeze-drying. During the freeze-drying process, the primary drying temperature is kept below Tg′ to ensure that the product remains in a glassy state. This prevents the collapse of the lyophilized cake and maintains the structural integrity of the matrix, which is crucial for the protection of proteins [33,34,47,48]. The temperature is often raised slightly during the secondary drying phase to remove bound water. However, it still needs to be below or just at Tg′ to avoid the transition to a rubbery state, which could destabilize the protein. Tg′ is the temperature at which a material transitions from a brittle, glassy state to a more rubbery, viscous state when in a partially frozen state. For freeze-dried formulations, the Tg′ value of the lyoprotectant is vital for maintaining the structural integrity and biological activity of proteins. A higher Tg′ provides a more stable glassy matrix that effectively traps proteins in their native conformation. This reduces the risk of protein unfolding or aggregation, which is often irreversible and leads to loss of function. In contrast, if the temperature rises above Tg′, the increased molecular motion can lead to partial unfolding or misfolding of the protein, promoting aggregation or the formation of inactive or harmful species [48,49,50].

Furthermore, during freeze-drying, water is removed from the system, which can lead to the denaturation or aggregation of proteins due to the loss of hydration shells and the stabilization provided by water [51]. Lyoprotectants with water-replacing ability can substitute for the role of water by forming hydrogen bonds and other stabilizing interactions with the protein. This helps to preserve the native structure of the protein in the dry state [34,47,48,52]. Proteins have hydration shells that prevent aggregation by keeping protein molecules separate. When water is removed during freeze-drying, these shells can collapse, leading to protein aggregation and loss of function. Lyoprotectants, such as sugars (e.g., sucrose, trehalose), can mimic the hydration shells and provide a protective coating around the protein molecules, preventing their aggregation and maintaining their solubility in the dried state. Disaccharides like sucrose and trehalose are classic examples of water-replacing lyoprotectants. They can form hydrogen bonds with proteins, replacing the stabilizing effect of water [46,52,53]. Compounds like glycerol and sorbitol also provide water-replacing effects and stabilize protein structures and some amino acids (e.g., glycine, proline) can act as lyoprotectants by stabilizing protein structures through water-replacement interactions. By mimicking the role of water, lyoprotectants protect against denaturation, aggregation, and degradation, ensuring the long-term stability and effectiveness of the dried biological product [39,41,53]. At low water content, monosaccharides and disaccharides have the capacity to create a glassy state. The molecules are immobilized and have minimal movement in this state. By limiting molecular motions and acting as “crowding agents” that lessen the probability of proteins coming into close contact and aggregating, these can aid in maintaining the native structure of proteins. Surfactants can stabilize the interfaces between the solid and liquid phases in the freeze-dried matrix, preventing aggregation [39,41]. A study showed Tween-80 as a potential lyoprotectant since it is a non-ionic surfactant that can disperse the *bacilli* and is able to prevent aggregation while rehydrating the lyophilized *bacilli* [54]. Many freeze-dried products are hygroscopic, meaning they readily absorb moisture from the air. Lyoprotectants help create a stable, glassy matrix that minimizes moisture uptake, preventing the product from becoming sticky or degrading due to moisture absorption.

### 3.4. Influence of Lyoprotectant on Structural Stability of the Lyophilized Product

Lyoprotectants play a critical role in contributing to the structural stability of freeze-dried products, which is essential for the preservation of the biological activity, integrity, and overall quality of the product. The removal of water can lead to the collapse of the freeze-dried cake, resulting in a dense, non-porous structure. This collapse can adversely affect the reconstitution time and solubility of the product [31,34]. Lyoprotectants help form a stable, amorphous glassy state that prevents structural collapse. A well-formed, porous structure improves the ease and speed of rehydration, ensuring that the freeze-dried product can be readily dissolved and used. A stable glassy matrix provided by lyoprotectants reduces molecular mobility, which in turn reduces the rate of degradation reactions [33,34,52]. Structural stability provided by lyoprotectants ensures that the freeze-dried product has a high surface area and appropriate porosity. This results in faster and more complete reconstitution when the product is rehydrated, which is especially important for pharmaceuticals and vaccines where rapid preparation is needed [55,56,57]. A stable lyophilized product should have an esthetically pleasing appearance, typically a uniform, porous cake. A collapsed or cracked cake can be a sign of instability and may affect the acceptability of the product by users or regulators. A well-structured, stable, freeze-dried product enhances the perception of quality and reliability among healthcare professionals and patients. The structural stability provided by lyoprotectants is essential to ensure the efficacy, safety, and longevity of freeze-dried products. By preventing collapse, maintaining protein integrity, and enhancing the reconstitution properties, lyoprotectants play a critical role in producing high-quality, reliable, freeze-dried pharmaceuticals, vaccines, and other biological products [37,50,57].

Structural integrity of the lyophilized product plays a major role in ensuring the protein stabilization. Achieving “uniform and elegant” cake appearance, amorphous nature, large porosity, and better solubility are the most important criteria to be met to produce a stable lyophilized product. Unstable cake commonly has a visual attribute of cracking, shrinkage, slanted cake, droplet formation, bubble, melt-back, fogging, and a puffed surface. Results revealed several lyoprotectants that contributed to better cake formation. A mixed lyoprotectant composed of sucrose, trehalose, and mannitol was added to mRNA-LNPs upon lyophilization and exhibited ginger root-shaped rigid structure with large porosity, good particle size distribution, encapsulation rate, and mRNA integrity [14]. Another study showed successful drying of *Bacillus Calmette Guerin* (BCG) with various sugars such as lactose, trehalose, and dextran-40; also, with sucrose and glucose at 5% to 10% concentrations, no adverse reactions such as shrinkage, melting, puffing, or collapsing were observed [19]. The addition of Tween-80 (0.04%) to the formula improved porosity, reduced bacterial aggregation, achieved minimal moisture (0.4%), and enhanced water reabsorption [19]. Moreover, trehalose added to the lyophilized PRV *rHN1201TK-/gE-/gI-/11k-/28k-* vaccine possessed an amorphous structure with a consistent porous interior which facilitated the movement of water vapor from the interior of the product to the surface, enhancing the efficiency of the drying process [17]. A variety of monosaccharides, disaccharides, trisaccharides, and amino acids were used as lyoprotectants, and disaccharides exhibited mostly amorphous natures with good protein stabilization. Monosaccharides such as Arabinose and Xylitol were amorphous, meanwhile mannitol remained crystalline with good cake appearance and stability. Monosaccharides were discouraged for long-term storage due to reducing sugar properties. The trisaccharide maltotriitol displayed good protein stabilization, amorphous nature, and was acceptable to good cakes, making it an effective lyoprotectant [21]. The antibody trastuzumab lyophilized with trehalose formed a cake-like structure and retained hydrodynamic size and charge (zeta potential) [25]. In addition, antibody rituximab freeze-dried with a combination of trehalose and mannitol ensured acceptable colloidal stability of GNRs and a uniform, spongy, and easily reconstituted cake. Trehalose proved to be the best excipient to maintain colloidal stability of GNRs while mannitol was the best cake-forming excipient [26].

### 3.5. Influence of Combined Lyoprotectants on Protein Stabilization

Another vital aspect that we explored was the combined application of lyoprotectants and the influence on protein activity and structural stability. A comprehensive protection achieved through the combination of various excipients not only stabilizes proteins during freeze-drying but also addresses challenges related to transport and storage [58,59]. Recent research emphasizes the use of combinations of lyoprotectants to optimize stabilization. For instance, combinations of sugars, amino acids, and polymers are used to leverage the different stabilization mechanisms of each component. The goal is to create a synergistic effect that provides maximum protection. Currently, advances in personalized medicine and precision drug formulation are driving the need for lyoprotectants that are specifically tailored to individual proteins or drugs. This precision approach aims to optimize the stability of each formulation uniquely. In a combined manner, sucrose and trehalose exhibit better shielding effects than individual application. Moreover, sucrose or trehalose co-formulated with other types of sugars, surfactanst, amino acids, or polyols has given positive implications on various protein modes. The mRNA-lipid nanoparticles (LNPs) lyophilized with sucrose/trehalose/mannitol mixture were stable at 2–8 °C, and they did not reduce immunogenicity in vivo or in vitro with a rigid structure with large porosity, which tolerated rapid temperature increases and efficiently removed water [14]. In a study about the lyophilization of premixed COVID-19 diagnostic RT-qPCR, they found that the sucrose/trehalose mix as protective agents exhibited effective performance at elevated temperatures [24]. The combination of sucrose and trehalose is encouraged due to their dual function as both cryo- and lyo-stabilizers. This evidence supports the fact that disaccharides are potential lyoprotectants as they protect protein integrity in high temperatures. This mixture maintained the enzymatic activity of lactate dehydrogenase (LDH) by 50% compared with the control [29]. One of the studies used a combination of lactose, sodium glutamate, Dextran-40, and tween-80 as a lyoprotectant. Lactose acted as vitrifying agent, MSG boosted the glass transition temperature (Tg), Dextran-40 as caking agent, and Tween-80 helped to disperse *bacilli* and prevent aggregation during rehydration [19]. The significant role of every lyoprotectant helped to stabilize the protein [19]. Moreover, an optimum freeze-drying formula could be achieved by a combination of trehalose and mannitol to ensure acceptable colloidal stability of GNRs and a uniform, spongy, and easily reconstituted cake. Biological activity of the conjugated monoclonal antibody after freeze-drying using optimized formulation was preserved [27].

### 3.6. Influence of Calcium Ion, Antifreeze-Glycopeptide, and Antiplasticized Polymeric Glasses as Lyoprotectants on Protein Stabilization

The earliest lyoprotectants were simple sugars, with sucrose and trehalose being among the most commonly used. Studies from the 1960s and 1970s recognized the stabilizing effects of these sugars on biological materials. Other disaccharides, like lactose and maltose, also began to be used as lyoprotectants. The use of these sugars was largely empirical, based on their availability and their effectiveness in empirical studies of freeze-dried products. The use of synthetic polyols and polymers like polyvinylpyrrolidone (PVP) and polyethylene glycol (PEG) emerged in the late 1980s and 1990s. These polymers offered new mechanisms of stabilization by enhancing the viscosity of the surrounding medium, which helped in reducing protein aggregation and degradation [39,54,60,61]. From the 1990s to the 2000s, the discovery of amino acids highlighted their role in both cryoprotection and lyoprotection. The discovery of naturally occurring antifreeze proteins (AFPs) and glycopeptides in the 1990s opened new avenues for protein stabilization. These compounds, found in cold-adapted organisms, provided insights into how biological systems can be protected at subzero temperatures. Polysaccharides like dextran started to be used in the late 2000s more frequently due to their excellent film-forming abilities and compatibility with various proteins. The use of cyclodextrins, which are cyclic oligosaccharides, became more prevalent due to their unique ability to form inclusion complexes with proteins, providing stabilization. Other than common lyoprotectants, a few lyoprotective agents were identified via screening, such as calcium ions, antifreeze glycopeptide analogues (GAPP), and antiplasticized polymeric glasses. Interestingly, one of the studies highlighted the use of Ca^+^ ions as protective agents of *Lactiplantibacillus plantarum*. The results showed that an addition of 0.50 mmol/L Ca^2+^ to the culture medium significantly improved the freeze-drying survival rates of the strain (*p* < 0.05). It was found that Ca^2+^ significantly increased the content of surface proteins. Due to the enhanced cell wall proteinase (CEP) activity, the surface proteins were anchored to the cell wall, thereby reducing the cell wall damage in strain during the drying process [12]. As a protein stabilizer, antifreeze glycopeptide analogues (GAPP) provided new insight. After non-enzymatic glycation, the glass transition temperature of GAPP rose to 21.4 °C, up from 27.6 °C. GAPP could significantly prevent the activities of galactosidase and lactic dehydrogenase from decreasing and prevent the leakage of proteins and nucleic acids within the cell. Cells disruption and contents leakage could be observed in control groups, while cells with GAPP maintained satiation and integrity [30]. Antiplasticized polymeric glasses are another interesting lyoprotectant discovered during the screening. This study confirmed that FMD (ficoll/maltitol/DMSO) and DMD (dextran/maltitol/DMSO) systems maintained ~90% activity compared with fresh extract at 4 °C for 3 months. At 25 °C, lyoprotectants were more effective than at 4 °C; DMD and FMD samples retained over 36% activity. FMD-protected samples had >30% full extract activity at 37 °C after 3 months, around 40% better than the extract-only control. FMD demonstrated the best protection at 50 °C for the first month [20].

### 3.7. Influence of Dissaccarides on Protein Stabilization

In this review, 15 out 19 screened articles had selected sucrose or trehalose or both sucrose and trehalose as lyoprotectants. These non-reducing disaccharides are well known and predominantly used as protein stabilizers due to their magnificent ability to replace water content in the protein model. This phenomenon is explained through water replacement theory stating that sugars substitute water molecules in their interactions with the polar groups of membrane lipids. The main gel of the membrane does not rise to the temperature of the fluid phase transition due to these interactions, which preserve the distance between lipids. Consequently, dry membranes do not undergo a phase shift during rehydration and maintain their fluid condition at physiological temperatures [6,33,50]. These sugars are able to interact with the protein and form hydrogen bonds to maintain their tertiary and secondary structures, which are critical for protein stability and activity [31,32].

The hydroxyl groups (-OH) of disaccharides create hydrogen bonds with protein functional groups. A study conducted by S. Dean Allison and team concluded that hydrogen bonding of disaccharides directly to dried lysozyme, and not the trapping of water by the sugar, prevents the unfolding caused by freeze-drying stress [33]. Presence of sucrose or trehalose improved the enzyme activity and achieved an 80% release of loaded activity of lyophilized Vegetal diamine oxidase (vDAO) [15]. However, another study that investigated the stabilizing effects of trehalose by comparing with sucrose concluded that trehalose stabilized theprotein myoglobin (Mb) better than sucrose. The results justified that the rotational motion around dihedrals between the two glucose rings of trehalose was slower than around the dihedrals between the glucose and fructose rings of sucrose. The faster dihedral rotation of sucrose is likely to induce motions of the protein backbone, which, in turn, leads to destabilization [18]. To support this, the *Bacillus Calmette Guerin* (BCG) strain that was lyophilized with trehalose showed a longer shelf-life with 12% heat stability (28 days for 37 °C), while sucrose exhibited declining activity of the strain. Surprisingly, lactose outperformed sucrose by achieving greater shelf-life and heat stability because humectant sugars such as lactose and trehalose with sufficient flexibility replace the hydrogen bonds and also have a higher glass transition temperature (Tg) to remain in glassy state during storage; they usually have more water binding capacity and have better protectivity than crystalline sugars such as sucrose [19]. Sucrose incorporation into freeze-dried trivalent antivenom (FDTAV) has maintained the protein concentration with 80% immunoglobulin activity, with no visual alteration in turbidity and no aggregates over the time [22].

## 4. Limitation and Challenges

Conducting a systematic review on the effectiveness of lyoprotectants in protein stabilization during lyophilization poses numerous challenges and restrictions at different stages of the review process, such as data search, screening, and analysis. These problems can have an impact on the review’s results’ strength and dependability. This section discusses the limitations and challenges faced from the initial to final stages of the review preparation.

Firstly, there is a limited availability of specific studies. Although lyoprotectants have been extensively studied, there is a shortage of research specifically focusing on proteins or combinations of lyoprotectants. This scarcity in focused studies limits the available data for a thorough and comprehensive analysis. Moreover, niche applications or specialized protein–lyoprotectant combinations may be underrepresented, leading to potential knowledge gaps. Different studies often use varied terminologies when describing lyoprotectants, protein stabilization mechanisms, and lyophilization methods. These inconsistencies can make it difficult to conduct systematic searches and may result in the exclusion of relevant studies. The lack of standardized terms may also complicate the comparison and synthesis of data from different sources.

Furthermore, language barriers are a vital limitation. Most systematic reviews limit their scope to English-language articles, which can exclude valuable research published in other languages. This can lead to geographical and cultural biases in the data, as important studies from non-English-speaking countries might be left out of the analysis, not excluding the fact about the heterogeneity of studies. The studies on lyoprotectants vary greatly in terms of their design, lyoprotectant concentrations, types of proteins, drying protocols, and evaluation methods. This diversity introduces a level of complexity that hinders the ability to conduct meta-analyses or draw generalized conclusions. The differences in experimental setups make it difficult to compare outcomes across studies reliably.

In addition, many studies report qualitative outcomes, such as visual assessments of cake structure or protein aggregation, rather than quantitative measures like residual moisture content or Tg′ values. This discrepancy between qualitative and quantitative data complicates efforts to synthesize results and perform statistical analyses. It also makes it harder to measure the overall effectiveness of lyoprotectants based on objective metrics. Incomplete data are another limitation that is vital. Some studies do not report all the necessary data points needed for thorough analysis, such as specific lyoprotectant concentrations, storage conditions, or the statistical significance of findings. Missing data pose significant challenges when trying to assess the efficacy of lyoprotectants or compare results across studies, thereby hindering comprehensive assessments.

One of the challenges in conducting systematic reviews is identifying and removing duplicate studies, which is both time-consuming and essential to avoid data redundancy. Some studies may also have overlapping data or could be extensions of previous research, which can lead to redundancy in the analysis. Many studies on lyoprotectants fail to consistently report critical data, such as key outcomes, experimental conditions, and methods. This inconsistency makes it difficult to compare studies directly, complicating the synthesis of results and drawing conclusions about the overall effectiveness of different lyoprotectants. The rapidly evolving field is also a challenge faced by the authors. The field of lyophilization and lyoprotectant research is constantly evolving, with new techniques and formulations emerging regularly. This dynamic nature poses a challenge for maintaining an up-to-date systematic review, as new studies continuously add to the body of knowledge. Keeping pace with these developments while ensuring the relevance of the review is difficult. The effectiveness of lyoprotectants is highly context-specific, depending on various factors like the type of protein being stabilized, the specific lyophilization process used, and the environmental conditions during drying. This specificity can limit the generalizability of the findings, as results from one study may not be directly applicable to another set of proteins or conditions.

## 5. Conclusions

The research reviewed provides strong evidence that lyoprotectants with higher Tg′ values and effective water-replacement abilities significantly enhance protein stability and maintain structural integrity during lyophilization. Lyoprotectants with high Tg′ values, such as disaccharides (e.g., sucrose, trehalose), polyols (e.g., maltitol), and certain trisaccharides, provide a stable amorphous matrix that protects proteins from denaturation and aggregation. The glassy state achieved by these lyoprotectants prevents molecular mobility, thereby reducing protein degradation. Higher Tg′ values correlate with better retention of protein structure and activity, which is crucial for ensuring the therapeutic efficacy and shelf-life of biopharmaceuticals. Sugars like sucrose and trehalose can effectively replace water molecules, forming hydrogen bonds with polar groups on protein surfaces. This substitution stabilizes the protein’s tertiary and secondary structures, mitigating the stress induced by drying and freezing processes. Studies have shown that trehalose forms fewer hydrogen bonds with proteins than sucrose, resulting in less disruption of protein structure, highlighting the importance of the specific interactions between lyoprotectants and proteins. Achieving a stable and visually acceptable cake structure is vital for the quality of lyophilized products. A combination of lyoprotectants, such as sucrose, trehalose, and mannitol, has been shown to create uniform, porous, and spongy cake structures. These structures facilitate efficient rehydration and preserve protein activity. The addition of surfactants like Tween-80 can further enhance cake properties by improving porosity and reducing moisture content, leading to better protein protection. The exploration of less common lyoprotectants, such as calcium ions, antifreeze glycopeptide analogues (GAPP), and antiplasticized polymeric glasses, offers new insights into protein stabilization strategies. These agents have shown promising results in enhancing protein stability by increasing glass transition temperatures, reducing cellular damage during lyophilization, and preserving enzymatic activities under various storage conditions. The synergistic effect of combining different lyoprotectants, such as the mixture of sucrose and trehalose or the addition of mannitol, enhances the overall protective effect. These combinations can provide dual functionality as cryo- and lyo-stabilizers, offering comprehensive protection across a range of temperatures and conditions. In summary, the selection of lyoprotectants based on their Tg′ values, water replacement abilities, and compatibility with the target protein can significantly impact the stability and efficacy of lyophilized pharmaceutical formulations. Future research should continue to explore novel lyoprotectants and their combinations to optimize lyophilization processes and extend the shelf-life of protein-based therapeutics.

## Figures and Tables

**Figure 1 pharmaceutics-16-01346-f001:**
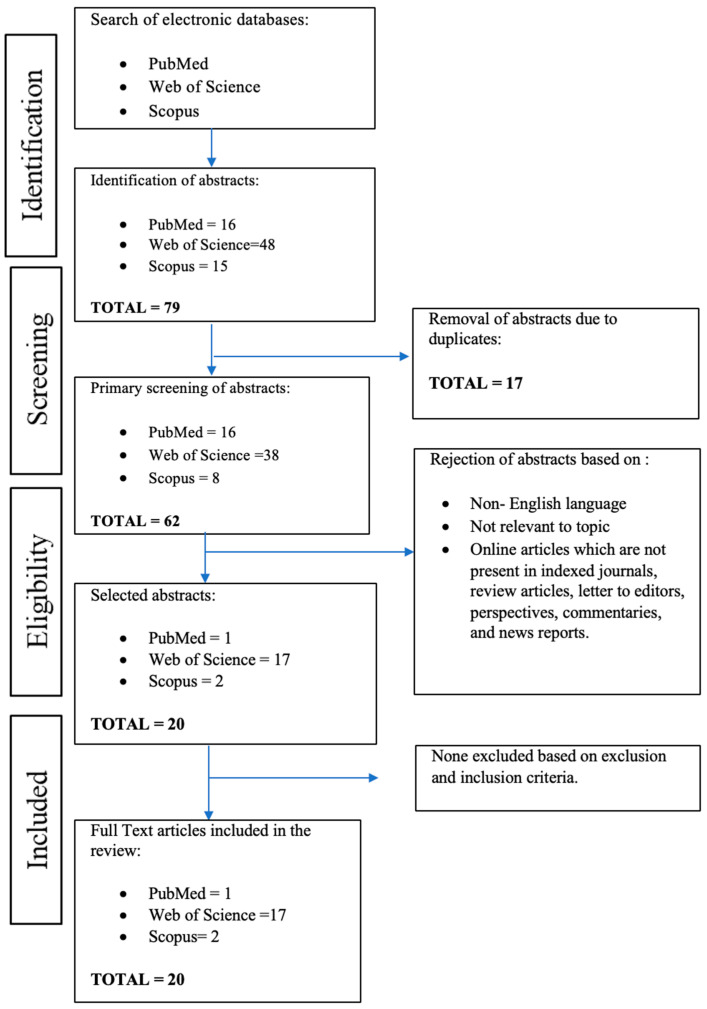
The PRISMA flow diagram of study selection.

**Table 1 pharmaceutics-16-01346-t001:** List of studies included.

Author	Lyoprotectants	Protein Model	References
S. H. Kim et al., 2023	Raffinose	*Bacillus* sp. *JY14*	[11]
S. H. Kim et al., 2023	Calcium ions (Ca+)	*Lactiplantbacillus*	[12]
J. Fortuin et al., 2024	Spirulina (Arthrospira platensis) protein isolate (SPI)	*L. rhamnosus GG* (LGG) cells	[13]
M. Li et al., 2023	Sucrose, Trehalose, Mannitol	mRNA-lipid nanoparticles (LNPs)	[14]
M. Megoura et al., 2023	Sucrose, Trehalose	Vegetal diamine oxidase (vDAO)	[15]
D. G. Farfan Pajuelo et al., 2023	Mannitol, Glucose	*Bacillus cereus* PBC strain	[16]
Y. Liu et al., 2024	Trehalose, Poly-vinylpyrimidine, Sorbitol	*TK/gE/gI/11k/28k* deleted pseudorabies	[17]
K. Ahlgren et al., 2023	Sucrose, Trehalose	Protein myoglobin (Mb)	[18]
M. Z. Nafchi et al., 2018	Sucrose, Trehalose, Glucose, Lactose, Dextran-40, Monosodium glutamate (MSG)	*Bacillus Calmette Guerin* (BCG)	[19]
K. M. Wilding et al., 2019	Ficoll/maltitol/DMSO (FMD), Dextran/maltitol/DMSO (DMD), Dextran	*E. coli* extract	[20]
T. P. Holm et al., 2021	Amino acids, Monosaccharides, Disaccharides, Trisaccharides (List of excipients: Table 2)	α-amylase, α-lactalbumin, albumin from human serum, albumin from human serum (fatty acid free, globulin free), β-lactoglobulin, bovine serum albumin, deoxyribonuclease I, γ-globulin, lactase, lysozyme, myoglobin	[21]
I. Mendonça-da-Silva et al., 2017	Sucrose	Trivalent Antivenom (AV)	[22]
C. Bou-Mitri and S. Kermasha, 2018	Dextran 6kDa, Sucrose, Mixture of Sodium benzoate, Potassium sorbate (BSKS), and Sorbitol	Laccase from *Coriolus hirsutus*	[23]
M. J. Hammerling et al., 2021	Sucrose, Trehalose, Dextran, Sucrose-Trehalose mix	GoTaq RT-qPCR kit	[24]
W. Ngamcherdtrakul et al., 2018	Trehalose	Antibody (trastuzumab)-conjugated mesoporous silica nanoparticles decorated with a copolymer of polyethylenimine and polyethyleneglycol	[25]
M. A. Hamaly et al., 2018	Mannitol, Trehalose, Combinations of trehalose/mannitol	Antibody rituximab	[26]
S. Darniadi, I. Ifie, P. Ho, and B. S. Murray, 2019	Maltodextrin (MD), Trehalose, Whey protein isolate (WPI)	Anthocyanin	[27]
V.-V. Auvinen et al., 2019	PEG 6000, Trehalose	Hepatocellular (HepG2) cell spheroid	[28]
A. Arsiccio et al., 2019	Sucrose, Trehalose, Cellobiose, b-cyclodextrin, Mannitol, Histidine, Sucrose-Trehalose	Lactate dehydrogenase (LDH)	[29]
X. Chen and S. Wang, 2018	Antifreeze glycopeptide analogues (GAPP)	*Streptococcus thermophilus*	[30]

**Table 2 pharmaceutics-16-01346-t002:** The classification of identified lyoprotectants.

Classification	Lyoprotectants
Monosaccharides	B-D-(+)-Allose, D-(−)-Arabinose, DL-Arabinose, L-(+)-Arabinose Dihydroxyacetone, D-(−)-Fructose, L-(−)-Fucose, D-(*)-Galactose, D-Glucose, D-(+)-Glucose, D-(+)-Glucose MH, D-(−)-Lyxpse, D-(+)-Mannose, N-acetyl-D-glucosamine, N-acetyl-D-neuraminic acid, D-Psicose, L-Rhamnose, L-Rhamnose MH, -(−)-Ribose, L-(−)-Sorbose, D-(−)-Tagatose, D-(+)-Xylose, DL-Xylose, D-(+)-Arabitol, L-(−),Arabitol, Erythritol, Inositol, L-Iditol, D-(−)-Mannitol, Ribitol, D-Sorbitol, D-Threitol, L-Threitol, Xylitol
Disaccharides	D-(+)-Cellobiose, 3-Gentiobiose, Isomaltulose MH, Lactose AH, Lactose MH, B-Lactose, Lactulose, Maltose MH, Maltulose MH, Melibiose, Palatinose hydrate, Sucralose, Sucrose, Trehalose, D-(+)-Turanose, Isomalt, Lactitol MH, Maltitol
Trisaccharide	1-Kestose, Maltotriose, D-(+)-Melezitose MH, D-(+)-Melezitose hydrate, D-(+)-Raffinose PH, Maltotriitol
Amino acids	Alanine, Arginine, Glycine, Lysine, Proline, Serine
Solubilizers	b-cyclodextrin, PEG 6000
Polysaccaride	Poly-vinylpyrimidine, Dextran-40, Dextran 6kDa, Dextran
Surfactants	Tween-20
Ions	Calcium ions (Ca+)
Chemical additives	Mixture of Sodium benzoate, Potassium sorbate (BSKS), Monosodium glutamate (MSG)
Antiplasticized polymeric glasses	Ficoll/maltitol/DMSO (FMD), Dextran/maltitol/DMSO (DMD)
Protein isolates	Spirulina (Arthrospira platensis) protein isolate (SPI), Whey protein isolate (WPI)

**Table 3 pharmaceutics-16-01346-t003:** The types of interaction between lyoprotectant–protein.

Interactions	Lyoprotectant	Mechanism	References
Hydrogen bonding	Trehalose Sucrose Mannitol (Sugars)	Form hydrogen bonds with the protein’s polar groups, such as the amide groups in the peptide backbone and the hydroxyl groups in the side chains. Stabilize the protein’s structure and prevent denaturation during the drying process.	[21,22,24,25]
Hydrophobic interactions	PEG (Polyols)	PEG interact with the protein’s hydrophobic regions, such as the non-polar amino acid residues. These interactions can prevent protein aggregation and maintain the protein’s native conformation.	[21,28]
Electrostatic interactions	Arginine Lysine (Amino acids)	Arginine and lysine can interact with the protein’s charged amino acid residues, such as aspartic acid and glutamic acid, in order to prevent denaturation during the drying process.	[21]
Osmoprotectant	Trehalose and Sucrose	Protect proteins from osmotic stress during the drying process and maintain the protein’s hydration state by preventing water loss during the drying process, which can cause protein denaturation.	[21,22,24,25]

**Table 4 pharmaceutics-16-01346-t004:** List of lyoprotectant and Tg′ values.

Class of Excipients	Lyoprotectant and Tg′ Value	Application as Lyoprotectant
Monosaccharides	D-(−)-Arabinose: ~−45 °C, DL-Arabinose: ~−45 °C, L-(+)-Arabinose: ~−45 °C, D-(−)-Fructose: ~−41 °C, D-(+)-Galactose: ~−36 °C, D-Glucose: ~−42 °C, D-(+)-Glucose: ~−42 °C, D-(+)-Mannose: ~−36 °C, L-Rhamnose: ~−37 °C, D-(−)-Ribose: ~−39 °C, D-(−)-Tagatose: ~−36 °C, D-(+)-Xylose: ~−46 °C, DL-Xylose: ~−46 °C, Inositol: ~−30 °C, D-(−)-Mannitol: ~−38 °C, D-Sorbitol: ~−32 °C, Xylitol: ~−35 °C, Glucose: ~−42 °C, Fructose: ~−41 °C, Mannose: ~−36 °C	Monosaccharides usually have lower Tg′ values and are often used in combination with other excipients.
Disaccharides	Lactose AH (Anhydrous): ~−28 °C, Lactose MH (Monohydrate): ~−31 °C, Sucrose: ~−32 °C, Trehalose: ~−27 °C, Maltose: ~−34 °C	Disaccharides are commonly used as lyoprotectants due to their relatively high Tg′ values compared with monosaccharides, which help maintain stability in the glassy state.
Trisaccharide	Raffinose: ~−35 °C	Trisaccharides like raffinose have Tg′ values that are somewhat intermediate between mono- and disaccharides, providing different stabilization characteristics.
Polysaccaride	Poly-vinylpyrimidine: 100 °C to 175 °C, Dextran-40: −15 °C to −20 °C, Dextran 6 kDa: −25 °C to −30 °C, Dextran: −10 °C to −20 °C (Tg′ increases with molecular weight due to reduced molecular mobility.)	PVP has a high Tg′ and is excellent for forming stable glassy matrices, providing good protection for proteins during freeze-drying. Dextrans have Tg′ values that vary with molecular weight, with higher molecular weights generally having higher Tg′ values.
Amino acids	Glycine: ~−35 °C (when used in combination with other excipients), Alanine: ~−45 °C, Proline: ~−52 °C	Amino acids generally have lower Tg′ values and are often used in combination with other stabilizers to achieve optimal protective effects.
Solubilizers	b-cyclodextrin, PEG 6000: −55 °C to −65 °C.	β-Cyclodextrin does not have a typical Tg′ value but can form glassy mixtures with other excipients. PEG 6000 has a relatively low Tg′, making it less effective as a sole lyoprotectant but useful as a plasticizer.

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
