# Peer review of "Effectiveness of Lyoprotectants in Protein Stabilization During Lyophilization"

_pharmaceutics, 2024, doi:10.3390/pharmaceutics16101346_

Round 1
Reviewer 1 Report
Comments and Suggestions for Authors
The topic of this manuscript is very interesting and intriguing and I appreciate the authors’ efforts of making a review on this topic. On the other hand, the manuscript is superficial and lacking details – it looks like rather a digest than a comprehensive review. Even though “Discussion” section provides some insights they are insufficient to make a good picture of the area, how it is developing, what are the challenges etc. I would recommend publication after a major revision.
Sentences on lines 53 and 57 are almost identical. One of them can be eliminated.
Methodology. It is very nice that the process of literature search is described explicitly. In my opinion, however, it is paid too many details. There is no need in Table 1 – everything is well explained in the text. Similarly, there is no real need in Tables 6 and 7. The main objective of the review to figure out which lyoprotectants do the job in the best way and why, so it would be nice to cover as many as possible relevant publications without thinking about potential biases. The number of identified relevant publications is fairly small (79 papers), which can be easy read and analyzed by even one person. Some papers published before 2019 can be potentially very relevant so there should be some search for such papers as well.
Table 3. Abbreviations should be explained. Lactose MH and a-Lactose MH (a = alpha) is likely the same compound.
Line 143. “Table 3” should be changed to “Table 4.” There is a reference to Figure 2 but no such figure is present in the manuscript.
Mechanisms of lyoprotectants action should be explained in more details. Otherwise, the Table 4 is just some sort of declaration without evidence and any science in it.
It is not very clear what stands for Tg and Tg’ and what relation do they have to glass transition temperatures (some explanations are given only at the end of the manuscript).
Line 177. “Table 4” should be changed to “Table 5.”
It would be nice to see how much protein stabilization can be achieved and how much this depends on the lyoprotectant concentration (especially taking into account the title of the manuscript).
Line 470. “Table 5 and Table 6” should be changed to “Table 6 and Table 7.”
Reviewer 2 Report
Comments and Suggestions for Authors
This manuscript reviews lyoprotectants used in lyophilization of proteins. Lyoprotectants are important in stabilizing proteins in lyophilization processes and thus the topic is important. However, the organization of the manuscript should be improved to clearly help readers understand the effectiveness and mechanisms of lyoprotectants as well as the most recent development in this field. Specific comments:
1. This manuscript is a review article. It is not clear if the manuscript should contain results and discussion sections as a research article. Possibly due to this organization, there are a lot of redundant statements. It would be suggested to revise the organization. Section 3 Results and Section 4 Discussion may be combined and divided into several sections with each section on one subtopic of lyoprotectants.
2. Background information on lyophilization and the need for lyoprotectants (now scattered in section 1, 3 and 4) can be integrated into section 1 Introduction or be described in a new section. Define Tg and Tg’ and state the difference between these two glass transition temperatures.
3. Classification and general mechanisms of lyoprotectants can be discussed in one section (now in section 3 and 4). The general mechanisms include the effects of lyoprotectants on Tg’, water replacement, structural stability, and bonding.
4. Each class of lyoprotectants and specific mechanisms can be discussed in separate sections.
5. Other sections follow e.g., limitation and challenges, conclusions.
6. Tables can be added to each section accordingly.
Reviewer 3 Report
Comments and Suggestions for Authors
Review article is well organized. However, the articles search was conducted from 2019, which is fairly recent and may not reflect the complete picture of the authors objective. Therefore, if the search can go back to at least 10 years (2014), that can improve the overall importance of this article and interest to the readers as well as researchers.
Round 2
Reviewer 1 Report
Comments and Suggestions for Authors
The authors substantially improved the manuscript. I think it can be published in its current form although I still do not like the overall approach of considering publications for five last years only and somehow superficial nature of the review.
Reviewer 2 Report
Comments and Suggestions for Authors
n/a